# A capability perspective on sustainable employability: A Dutch focus group study on organizational, work and personal conversion factors

Jan Meerman[1,2]*, Patricia A. J. van Casteren[1,3], Evelien P. M. Brouwers[1], Arno van Dam[1,4], Jac J. L. van der Klink[1,5]

1 Tilburg School of Social and Behavioural Sciences, Tranzo, Tilburg, Netherlands, 2 Dimencegroep, Zwolle, Netherlands, 3 Ascender, Utrecht, Netherlands, 4 GGZ Westelijk Noord-Brabant, Halsteren, Netherlands, 5 North West University, Optentia, Vanderbijlpark, South Africa

* j.meerman@dimencegroep.nl

## Abstract

### Objective

In the field of work, there is a shift towards more value-based approaches to study the sustainable employability of the present-day worker. The capability approach offers a value based and innovative conceptualisation and framework of sustainable employability characterized by contextuality, normativity and diversity. The capabilities of Dutch employees have been established and validated, yet it is not known which conversion factors on a personal, work and organizational level enable employees to achieve value in work in different Dutch occupational sectors.

### Methods

Our qualitative approach included seven focus groups in different occupational sectors including elderly care, higher education, insurance work, facility management and the oil-, car- and chemical industry. Each focus group included 5–11 participants and took approximately one and a half hour. A qualitative content analysis was used to analyse the data, by combining deductive and inductive coding respectively. Deductive coding involved assigning themes to the conversion of resources into capabilities at the organizational, work and personal level.

### Results

On the organizational conversion level, important themes were cultural aspects, power relations, shortage of personnel and policies for self-management. On the work conversion level, social contacts, communication and workload, tasks and schedules were identified. Social contacts were described as a work value in itself, but also conditional for achieving other work values. On the personal conversion level, experienced work stress, motivation and the ability to achieve values informally within the company.

**Data Availability Statement:** The data are held in a public repository: https://doi.org/10.34894/MLEDFZ.

**Funding:** The author(s) received no specific funding for this work.

**Competing interests:** The authors have declared that no competing interests exist.

## Conclusion

From our findings it follows that focus groups are sensitive to identify conversion factors on all three levels of conversion. In addition, companies and their employees might effectively increase work capabilities by being sensitive to all three conversion levels simultaneously. Further research is necessary to study the effect of a capability-based intervention at the work floor.

## Introduction

For the present-day worker, the *value* of work seems to be an important aspect of their quality of working life and sustainable employability (SE) [1–3]. In times where an estimated forty percent of workers worldwide perceive their jobs as irrelevant [4], it seems vital to study what is of value to workers to figure out how we could make their employment more sustainable. Jahoda was one of the first who conducted research on work values and highlighted income as the single most important work value [5]. Albeit, she considered other relevant values as 'latent values'. Nowadays, we acknowledge that indeed many other values such as relations, personal development and meaning play an important role [6]. In recent decades, values have been implemented in leading work and health models such as the Work Ability House of Ilmarinen [7] and the Job demands-resources model [8]. In his Work Ability House, Ilmarinen [7] has grouped values, attitudes and motivation as one important dimension impacting work ability of the employee. In the Job Demand-Resource model [8], recent developments have indirectly included personal values of the employee in studies by focussing on the person-job fit and job crafting [3, 9–11]. Other scholars have developed new conceptualizations and value based operationalizations of sustainable employability [12–14].

### Sustainable employability

In the literature there is no consensus yet on the conceptualization and operationalization of SE [14]. The most widely used definition is the definition proposed by van der Klink et al. [2], based on the Capability Approach (CA): "Sustainable employability (SE) means that throughout their working lives, workers can achieve tangible opportunities in the form of a set of capabilities. They also enjoy the necessary conditions that allow them to make a valuable contribution through their work, now and in the future, while safeguarding their health and welfare. This requires, on the one hand, a work context that facilitates this for them and on the other, the attitude and motivation to exploit these opportunities". Fleuren et al. [14] discuss several definitions of SE and propose, based on their research, SE means that "an individual's ability to function at work and in the labour market, or their 'employability', is not negatively, and preferably positively, affected by that individual's employment over time. This ability can be captured meaningfully as a combination of nine indicators (i.e., perceived health status, work ability, need for recovery, fatigue, job satisfaction, motivation to work, perceived employability, skill-gap, and job performance) that collectively describe how well an individual can be employed at different points throughout the working life." They state SE can be best approached as a longitudinal multidimensional individual characteristic. From their perspective, they criticize the CA-based definition of SE on several points, such as that the definition conflates causes and effects and there is unclarity on what the specific indicators of SE are and at which level SE should be captured (i.e., is it an individual characteristic or contextual). They

indicate that the relationship between individual and contextual factors in relation to SE needs to be unravelled. On this latter point, we follow their suggestion and this article aims to contribute to this aspect, yet from a capability perspective. On the other points, we want to make it clear that our position—based on the capabilities approach—is fundamentally different; we will discuss that below.

In essence, the CA starts from different conceptual and scientific principles [2], in comparison to positivistic research methods often used in the field of SE [12–14]. Central to the CA are normativity, diversity and contextuality. Normativity means on the one hand that the focus is not only on understanding and explaining relations between concepts and constructs, but also explicitly on ameliorating situations (in this case a flourishing and sustainable working life for workers). Another aspect of normativity is that individuals have entitlements on the context for being supportive and facilitating for achieving capabilities. With respect to diversity, Sen [15–18] (the founding father of the approach) has postulated the importance of on the one hand human diversity in how people flourish in life and on the other hand that people differ in their freedoms to achieve valuable activities and roles. Contextuality is important because the way in which the context facilitates and 'enables', partly determines the extent to which capabilities—and in our view SE—can be achieved.

In our view of the CA, there is less need for an exact position of constructs in the model—preferably linked by quantifiable relationships—and there is more room for a more flexible and realistic changing position of constructs in the model. This allows to identify–and eventually change—factors which in a specific context are necessary to convert resources into capabilities. It is therefore inherent in the capability view that constructs can be either cause or effect, an aspect of non-linearity, which is also seen in systems- and complexity theory [19]. Health, for example (an important construct for SE) can be either a resource or a capability (an achievable value in itself), but also a meta-capability (a condition for realizing other values, possibly a conversion factor). And health can also be a 'functioning', a realized outcome: being sustainable employable contributes to well-being and health.

For everyone, the set of opportunities one has for realizing important values and goals (the capability set) is central, although the content of those values/goals can vary as well as the context in which they must be achieved. From this perspective, scholars will first have to focus on the identification of the capability elements by means of qualitative methods such as interviews or participatory methods. Thereafter, one could objectify the capability elements and interactions between resources, conversion factors and functionings by means of methodological triangulation techniques such as member checking or repeated measures. Or one could focus which interventions impact capabilities of workers.

In this view, SE from a CA-perspective is always tailor-made for this person in this context. Two people with the same work abilities, possibly differ in the extent to which they can achieve their values, depending on individual preferences and facilitation (or limitation) of these values by the context. General indicators or determinants are often of little meaning in the specific work context in which SE has to be increased. A correlation of 0.23 says little in a particular situation (n = 1) where such determinant is or is not present. This study focusses on the identification of these conversion factors, or those contextual factors essential to convert his or her resources into capabilities. In short, our research question is: what conversion factors on an organizational, work and personal level enable employees in their specific context to achieve value in work in different Dutch occupational sectors?

As the reader might be unfamiliar with the capability approach and its application to the field of work, one is referred to the theoretical background following this introduction.

## Theoretical background

**The capability approach.** The capability approach (CA) [15–18, 20] is a value-based approach, which conceptualizes wellbeing in terms of *capabilities*. The CA emphasizes that not only resources or utility-outcomes matter to create equal opportunities for a flourishing in life, but more particularly the *freedom* to which humans can flourish in life. The CA is determined to be sensitive to human diversity and socio-cultural contexts [20]. In the CA, capabilities are the real freedoms one has for being who one wants to be (beings) and doing the things one has reason to value (doings). Capabilities sprout from the freedom that institutions, companies and social relations provide for individuals to convert resources into a set of tangible opportunities (capability set) that can be used to achieve various valuable achievements (*functionings*) [15–18]. Individuals and contexts differ in their ability to convert resources into capabilities. *Conversion factors* are those factors that impact on whether people are able and enabled to convert resources into the actual realizations of *functionings*. It is precisely because of conversion factors that the CA focuses on capabilities and not on resources or functionings [15, 21]. *Resources* (such as income, capacities and wealth) only have meaning, if they enable value realization. Inequality in capabilities between people might be due to lack of resources, but also by different levels of conversion factors. Robeyns [20] has made a general distinction between personal, social and environmental conversion factors. Personal conversion factors are internal to the person, such as age, gender, skills, ethnicity and disability. Social conversion factors include social norms and culture, ethnic profile, gender roles and power relations. Environmental conversion factors are climate and means of transportation.

**The capability approach applied to the field of SE.** In the field of work and health, van der Klink et al. [2] hypothesized that increasing capabilities (the freedom to achieve values in work) adds to employees' sustainable employability by making people more resilient and resistant to stress and hereby increase the wellbeing of employees. In Fig 1 the operationalization of the essential elements of the CA as described above are visualized in an adaptation of the model of van der Klink et al. [2] to the current study. From Fig 1 it follows that workers have resources or work inputs, such as personal capacity, task structure and work demands. Work

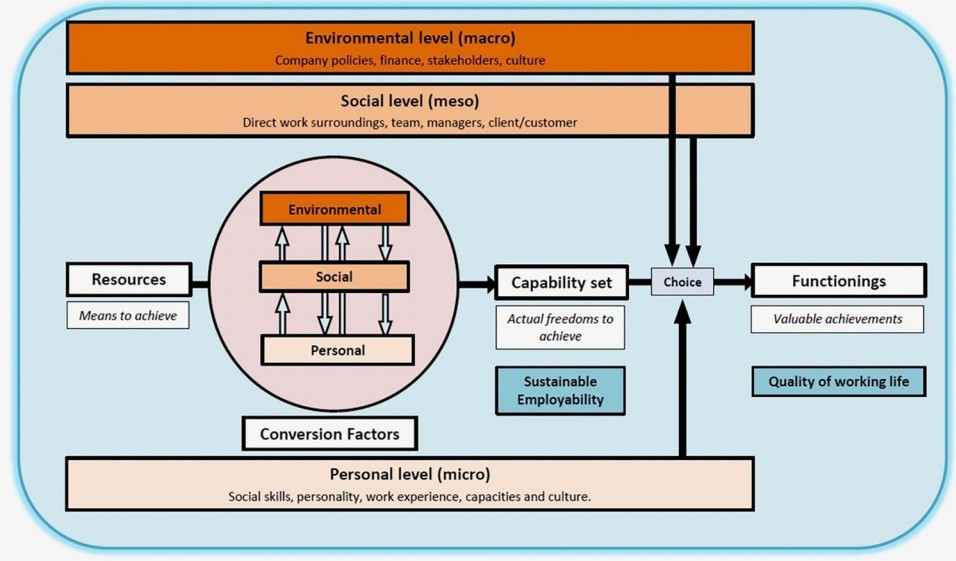

**Fig 1. Adapted version of the capability model applied to the context of work by van der Klink [2].**

inputs are not determinants of SE, they are factors that can lead to a set of potentials to achieve valuable work functioning, provided that appropriate conversion factors are present. Based on this model, seven work values were identified and validated in a representative population of Dutch employees [21]: using knowledge and skills, applying knowledge and skills, involvement in important decisions, building and maintaining meaningful contacts at work, setting own goals, having a good income and contributing to something valuable. For the development of the work capability set a combination of interviews and expert meetings was used. This capability set represents the sustainable employability of the worker in its work context [2], whereas functionings are regarded to be related to the current wellbeing or quality of working life. A value can be considered a capability when it is important to the individual worker, (s)he is enabled by the work context, and (s)he is able to achieve it. Sen has argued to establish capabilities based on a democratic process of critical scrutiny in order to respect human diversity and the voice of minorities [16]. Therefore, a combination of participatory research, interview and focus groups have been regularly applied in the capability literature [22–24].

**Conversion factors.**  The relationship between personal and contextual factors which convert resources into capabilities has not deserved much attention in the capability- and the work and health literature. Following Robeyns [20] and van der Klink et al. [2], it is assumed that in work, three levels of conversion are of importance. Organizational conversion factors relate to policies of the company concerning the work of the employee and include societal factors (such as legislation). Work conversion factors are related to the direct work surroundings of the worker which impact performing of daily work tasks. Personal conversion factors are related to the background of the participant applicable to the work context, such as social skills, capacity to perform tasks, activities and take certain roles, sociocultural values motivation and energy level. In work, for example, if one has the value of developing knowledge and skills, the way a company facilitates courses is an organizational conversion factor, the manager that allows time and financial resources to join the course are work conversion factors and the motivation to study and prepare the courses in private time are personal conversion factors. Without these conversion factors it is not possible to convert resources into capabilities.

## Methods

We chose focus groups as a methodology to identify conversion factors, because focus groups have the advantage of discussion between a diverse range of workers with different tasks and roles within a company. Furthermore, focus groups provide a room for dialogue, not only to identify capabilities, but also for reflection on achieving values of different people and what possibly facilitates and limits them in achieving values. Moreover, the focus groups might provide immediate suggestions for interventions for the group or individual workers. The consolidated criteria for reporting qualitative research (COREQ) were used to comprehensively report the focus group process [25].

### Participants

Focus group participants were recruited in different occupational sectors in order to include a broad range of the overall Dutch labour market. In this way, we aimed to include different work contexts, including organizational conversion factors. Direct contact persons of the researchers at seven Dutch companies in different occupational sectors were approached by phone. These contact persons were asked to sample a group of collaborating colleagues within the company, as it was assumed these colleagues would be better able than non-collaborating colleagues to discuss conversion factors at the work floor. Furthermore, individuals were eligible to participate when speaking Dutch fluently, were aged between 18 and 67 and interested

**Table 1. General characteristics of focus groups.**

| Description of group | Occupational sector | Number | Age (years: mean min—max) | Gender (m/f) | Highest attained Education (MBO/HBO/WO)* | Service years (mean min, max) | Work hours (part time/ full time) |
|---|---|---|---|---|---|---|---|
| Teachers and supporting staff | Higher education | 6 | 45,2 (31–60) | 2/4 | 1/1/3# | 6,9 (3–8) | 0/5# |
| Personal health care assistants | Eldercare | 6 | 46 (30–56) | 0/6 | 6/0/0 | 20,1 (3–33) | 6/0 |
| Facility service team | Mental health | 6 | 56 (39–63) | 3/3 | 6/0/0 | 25,1 (7–45) | 1/5 |
| Mixed (mechanics, personal services, occupational physician and managers) | Oil industry | 9 | 51,7 (36–58) | 7/2 | 4/2/3 | 14,4 (1–35) | 0/9 |
| Sales employees | Transport industry | 5 | 44,6 (28–60) | 5/0 | 1/4/0 | 11,3 (3,5–16) | 0/5 |
| Physicians and reintegration managers | Insurance | 6 | 44,5 (31–55) | 0/6 | 1/3/2 | 7,25 (50–11,5) | 2/4 |
| HRM-personnel | chemical industry | 11 | 38,1 (23–62) | 6/5 | 0/4/7 | 8,03 (1–28) | 1/10 |

#Missing data

*Dutch education system: MBO = medium vocational training, HBO is higher vocational training, WO = university degree

to participate in discussion on the topic of sustainable employability in their company. The groups needed to include a minimum of five participants. See Table 1 for the general characteristics of the focus groups.

## Procedure

The focus groups were conducted in the period from 2016–2017 in private meeting rooms at the company and lead by alternating combinations of two members of the research team. The moderators were all experienced in conducting interviews. The moderators did not know any of the participants before the focus groups. Beside the participants and the moderators, no other people were present during the interviews. Interviews took between sixty and ninety minutes. The focus groups were audio recorded.

Each participant filled in an informed consent before the start of the interview. The focus groups started with brief general information and structure of the interview. During the first five minutes, respondents were asked to silently reflect on what they valued in their own work, and to write this down. Work capabilities were provided on paper to start the discussion on conversion factors with a certain degree of uniformity on the work values [21]. Thereafter, moderators opened discussion to aggregate reflections on what factors allowed for work value realization for the group members. First it was discussed if participants recognized these work values in their current work. The focus of the interview was then put on which factors abled or disabled them in realizing those values. Participants were encouraged to talk openly about their experiences, opinions and values. The moderators aimed to divide attention across all participants as fairly as possible, to avoid unfair prominence of participants.

## Data analysis and synthesis

The focus groups were transcribed verbatim and anonymized and in succession analysed with the use of Atlas Ti version 7 for Mac. Participants did not receive any transcripts to give feedback or comments. Based on Braun & Clarke [26] a qualitative content analysis was used combining inductive and deductive coding [27]. At first, based on open coding, segments of the seven transcripts were coded by two authors independently (JM and PvC). Subthemes were

identified within the individual focus groups. After coding, both authors discussed similarities and differences in codes, which were resolved by discussion. Thereafter, themes were identified by comparing subthemes across focus groups. A framework of themes and subthemes was built. Thereafter, both authors divided the themes in one of three conversion levels, deductively, based on pre-established approach of conversion levels in de capability set for work model [2]. Both coding authors discussed the assignment of themes to the micro, meso or macro conversion levels. Subthemes, themes and conversion levels were discussion in the research team until consensus was reached. The themes are presented in the result section, with supporting citations by focus group participants. The quotations have been translated in English by a native speaker.

All employees participated on a voluntarily basis and received a gift voucher of 15 euros in return. The data were used for research purposes only. Prior to the study, approval was obtained from the Ethics Review Board (project number EC-2015.28) of the Tilburg School of Social and Behavioural Sciences. Informed consent was obtained from all participants prior to the interviews.

## Results

See Table 2 for an overview of the main findings. The themes regarding conversion factors on each conversion level are presented below, supported by citations. For practical reasons, focus groups will be either consistently referred to as the common function of the participants or the organizational sector.

### Organizational level

The employees in the oil industry- and HRM-group expressed their desire to have more local decision-making power to design their own work processes suitable to the Dutch context instead of direct translation of the company's global policies based on American law and policy.

> *"Sometimes I think that actually America should not interfere with ours. Let us do our own thing, then things would be a lot easier for us. Now we get universal regulations, which we as a team can't pursue here in the Netherlands."* (employee in HRM-group)

Employees from the insurance group and higher education group emphasized the current collaborative atmosphere between colleagues. These participants had experienced the shift

**Table 2. Overview of conversion factors identified on each level of conversion.**

| Conversion level | Conversion factors |
| --- | --- |
| Organizational (macro) | • Cultural aspects—social atmosphere, appreciation and support<br>• Power relations and decision making<br>• Policies for self-management<br>• Shortage of personnel |
| Work (meso) | • Social contacts at the work floor—colleagues, manager and clients<br>• Communication among collaborating colleagues or the team<br>• Workload, tasks and schedules |
| Personal (micro) | • Psychological experience of work stress<br>• Active or passive stands toward changing their personal work situation–via formal or informal ways<br>• Motivation for work activities (degree of challenge, diversity in work) |

from a competitive and male-dominated culture (in former employment), in which they did not feel comfortable among colleagues.

*"I even worked in the financial sector, for over ten years. Suddenly I just quit my job. I had not even applied for another job, but I was so done with it. Actually, I earned twice as much then, but I just very consciously chose to quit. And I am glad about it. It's the way we work together here, the way we interact with each other, have respect for each other, value each other, stimulate each other. . ."* (employee in education group)

*"I also worked for a bank, mortgage, credits, but that has always been with a lot of eh. . . males, which made it very impersonal really. And now we have this kind of family, we also see each other privately. You just can tell your personal stories to each other, and I would miss that a lot when I would leave, yes."* (employee in insurance group)

Employees of the oil industry group explained how the former punishment-based culture had deteriorated the work atmosphere among colleagues, which fuelled a fear of making mistakes. Changes towards an improved learning environment in the company has led to an environment in which participants feel more valued for the work they do.

*"For a long time, we had a shame-and-blame culture. In practice it meant that if employees did something wrong, they would immediately get a file note or an official warning, also from their direct colleagues. Nowadays there has been a small shift towards an environment in which learning is more important than punishment. That feels like a step in the right direction."* (employee in oil industry group)

In addition, in the car industry, HRM group, the insurance group and the oil industry group, the top-down management culture of the company was limiting bottom-up initiatives proposed by the teams or individuals. Many workers experienced managers blocking attempts to realize valuable ideas. As a result, they experienced a lack of appreciation and support.

*"The decision making often takes a very long time. {. . .} You have to cross a few management departments to get something done. So, in the case everyone around you is enthusiastic [red: about your plan] even up to two or three bosses above you, eventually something will go wrong along the line, nobody knows what, and the plan is dismissed in the end."* (employee in car industry group)

*"When we think of something with the team and we all become convinced that this idea is very important and very good for the company, but then it ends up somewhere in a drawer. (. . .) I sometimes miss the appreciation and recognition for the fact that employees can have very good ideas. (. . .) I am just not going to try it again, and that is of course no good at all."* (employee in insurance group)

In the HRM, education, insurance and car industry group, employees and teams could manage their own work tasks as long as production goals were achieved. These workers also mentioned they could manage their working hours in balance with the needs of the team and their private working life. Participants in the eldercare and facility service group reported less opportunities to manage their own work tasks and hours.

*"What I do find really important is that I was judged on the result of my work and not on the hours I worked. I think I really got the space to be flexible in this respect. Of course, you need*

*to be present at certain hours and times slots, I would just appreciate that remains the same in the future."* (employee in HRM-group)

*"You have to do a certain amount of work, you know when these things need to be done, but you can choose if you want to do the work in the morning or afternoon. And if there is an emergency, that is not a problem, you do that first. But you can in a sense sort out your own day with you own energy level in mind."* (employee in insurance group)

In the facility service and eldercare groups, ongoing cuttings in national healthcare were limiting education funds. These focus groups discussed that although many learning courses were available on paper, employees could not actually participate in these courses. As a result of a shortage of employees, daily work pressures had increased steeply. Due to ongoing challenges to complete work schedules, there was simply no time for additional courses, unless in private time.

*"We are obliged to do some courses, but past times we could not go because there were no other employees at the work floor. That is too bad, I really appreciate these courses."* (employees in facility service group)

*"Last year everyone had to fill in what courses you would like to follow. Well, we also did this three years ago, but nothing was done with this."* (employee in car industry group)

## Work level

For most groups, social contacts were interpreted as an inherently valuable factor of their work, yet in the insurance and the eldercare group, social contacts were also described as a work conversion factor to facilitate achievement of other work values, especially employee's contributions to create a proper product or good quality care for the team, customers, clients or the company.

In the eldercare group, employees were in many ways dependent on each other to provide quality care for the elderly. The physically strenuous work within the tight work schedules had led to a high workload which limited them to keep up high quality care for their patients, and less freedom to enjoy their work. However, the social contacts between these eldercare nurses were regarded by them to be of essential support to deal with the psychological burden of the work. Besides, they explicitly appreciated the value of working as a team to provide sufficient quality care for their customers.

*"Being with my colleagues, that is what I find important. We know each other. If something is up you can talk about it with someone else, about everything really. {. . .} The physical stress in this work is really high. But I think that the psychological pressure of our job is relieved, because we talk a lot about it with each other. We agree about many of the things we talk about, and therefore the pressure on our job reduces. If we would have had a team that was not as close as we are right now, maybe the psychological pressure would have been much higher."* (Employee in eldercare group)

In the insurance group, employees discussed their preferred choice to work as a team. The good social contacts with colleagues and clients in this team facilitated many of the work values discussed in this group on a daily basis.

*"Colleagues are important. That is something which I believe is working out really well now and with that you get a lot done, and short communication lines. (. . .) 'Every day we ask*

*ourselves in our team: we need to deliver good quality work and how are we going to achieve that as a team?" (employee in insurance group)*

In other groups such as the HRM and the oil group, employees did appreciate collaboration and contact with colleagues. In the experience of some of these employees, the absence of social contacts between colleagues sometimes negatively impacted the feeling of working pleasantly.

*"What I do find important are my colleagues, and to have a positive feeling about them. I have experienced fewer positive feelings. If you don't feel a click with your colleagues, it is impossible to work pleasantly."* (employee in HRM-group)

## Personal level

On the personal level, workers seemed to differ in how they contributed to the realization of their values in work. Many workers expressed their motivation to put energy in the job in order to create value in and recognition from their work.

*"I really want to learn new things, and what is also important, it is not only execute my job, but also putting this little extra into it. That I could really mean something more, to be of additive value, that gives me a great feeling, and that I go home happy, if I know I really meant something, that is what I think is important."* (employee in HRM-group)

*"You are there for the people and you try to make the work as agreeable as possible right? (. . .) If you have worked all day very hard and it is tough, but you hear at the end of the day that they are glad for the care you have given, that gives a good feeling, that's what you do it all for."* (employee in eldercare group)

Some workers had become more passive, often because they had learned within their work context that organization barriers could not be changed by their personal efforts, or efforts by their team or due to lack of variety and challenges offered in their job.

*"At a certain moment the job becomes routine for me and as a consequence, I am really done with it."* (employee in HRM-group)

One employee from the sales department of the car industry over the years had let go of his efforts to improve the efficiency of IT-systems, still it seemed he had accepted a lack of value realization with the sight of earning a proper salary up to his pension.

*"If that boss wants that I type in everything one by one, then I type everything one by one. I don't have a problem with that anymore. In the past I have been very annoyed by this, but now I laugh about it. (. . .) Hundred times the same task, it does not matter to me anymore. I have always worked my 8 hours a day. What I do in these hours is up to my boss or the firm, but I just carry out what they want me to do."* (employee in car industry group)

In contrast, other individual workers in the sales and the oil industry group discussed they purposely choose their ways around the formal policies of the companies by means of personal strategies and soft skills. These workers made use of meaningful contacts at work to conduct their work tasks. Moreover, their soft skills often contributed to the realization of something valuable.

*"But you really start looking which work you can do in which shift: the one shift is easier than the other shift. That is going to be nice! So, if you have more complex jobs, I take shift 4, and if you have easier jobs, I take shift 1, because if you put more complex problems in shift 1 you know for sure the entire permit is lost. That makes this work fun"* (employee in oil industry group)

*You hear most news in the smoking rooms. You don't have to smoke, but you must get in there, I need to hear what's going on. It may be unhealthy, but it is also just fun* (employee in oil industry group)

*"We have found our ways to do extra things, doing things right across everything, I did secretly choose my own path somehow."* (employee in insurance group)

Some employees in the insurance, HRM group and education group, stated that their work was so valuable to them, that they also worked in their spare time to work on interesting and challenging projects. They did not always seem to care their work tasks exceeded the normal boundaries between private time and work time.

*"I think I work about 50 hours a week. Yes, I do a lot of work in the evening, and in the weekends, but I don't see those things as work. I choose to do these projects myself, and I like these things, and I get a lot of freedom to get where I want to go."* (employee in insurance group)

*"About the E-learning I developed, I didn't have time for that, at all! But, I loved this initiative so much, I just worked through Christmas and another weekend, you just go for it."* (employee in higher education group)

## Discussion

### Key findings

The research question of this study was what conversion factors on a personal, work and organizational level enable employees to achieve value in work in different Dutch occupational sectors. Several themes on all conversion levels have been found across the focus groups. On the organizational conversion level, workers reported cultural aspects, power relations and shortage of personnel as important factors impacting value achievement. A strong focus on safety (chemical industry group) and/or work efficiency (elderly care group, insurance group), was accompanied by vertical hierarchies and a focus on top-down regulations. In some groups, employees stated that their ambitions and ideas were usually not recognized by the organization and therefore impacted their motivation to realize their values. On the work conversion level, social contacts among colleagues, communication and workload, -tasks and -schedules were identified as main conversion factors. It was remarkable that social contacts were described as a work value in itself for some employees, but also conditional for other employees for achieving other personal values such as personal work efficiency and personal development (see also discussion below). Social contacts with clients, colleagues or managers could be either facilitative or limiting value realization. On the personal conversion level, people differed in how active and motivated they were to achieve values within their current job. In addition, too much work stress seemed to affect the realization of work values. Some employees explicitly reported their ability to achieve values informally within the company, most often by applying their experience in the company and social skills.

## Normativity, contextuality and diversity

In the introduction we have introduced the CA based on normativity, contextuality and diversity. In this focus group study the main goal was to identify conversion factors in relation to work value realization. In terms of normativity, this focus group method seems also suitable to be used as a first step for an intervention study or action research to increase SE within a company. In relation to contextuality, this study seems to ascribe that conversion factors at multiple levels influence value achievement of the worker. As an example, it was seen that for each focus group limiting conversion factors could be on all three levels of conversion. In one group the organization could limit the value to make one's own decisions and plans, in another group, at the work level, the communication between colleagues could facilitate the application of knowledge and skills and on the personal level some employees seemed better in others to find informal ways to achieve values such as setting one's own goals. In terms of diversity, the findings suggest that specific characteristics seem important to realize values. This supports the need of tailored made analyses and interventions to support the diversity of workers working in the same work context. Instead of facilitating or limiting factors found on a group level which might be determinants of SE in a positivistic sense, conversion factors have been found which seemed essential in the process of value realization of employees in their specific situation.

## Social contacts as meta-capability

Social contacts seemed to be an important work value in itself, but when this capability was actually achieved, it also seemed to impact the achievement of other capabilities in some groups. These findings on social contacts support the concept of 'meta-capability' as proposed by Venkatapuram [28]: a capability such as health, is foremostly a capability in itself, but also an essential conversion factor in the realization of other capabilities. In health, the concept of meta-capability is inherent in the concept of health. For social support, we found that the situation determined whether social support was a meta-capability—namely if a team has or feels a joint team responsibility. In the eldercare group e.g., social contacts between colleagues were found to be a psychological buffer to live up to the high physical work demands, but also to achieve other values such as the creation of something valuable. These results are in line with earlier research that indicates that social support is an important buffer against job strain [29–32]. On the other hand, when social contacts are characterized by large communication and lack of recognition by managers or clients such as shown the facility service and eldercare group, social contacts can severely limit achievement of other values.

## The concept of value in relation to SE

In the introduction it was stated that values are increasingly important in the field of work and health and in SE. The results from this study offer preliminary support that values indeed deserve more attention in the conceptualization of SE. If high work demands fully align with the work values of the employee, they might become facilitative work conversion factors which increase SE, such as perceived by certain workers in the higher education group. On the other hand, when a large share of demands does not contribute to value realization, this seems to negatively impact the achievement of values such as in the facility management group.

## Strengths and limitations

Strengths of this study are the use of multiple focus groups in a broad variety of occupational sectors. The conceptual framework of the CA in combination with the chosen method proved

suitable for identifying conversion factors in the workplace. In most groups, especially organization- and work conversion factors were discussed thoroughly. Moreover, the CA has helped to structure the open coding in the data analysis adequately to identify themes of conversion factors on the three levels. By keeping a uniform set of work values [21], some important differences in conversion factors between workers and groups have become visible. As conversion factors are related to relevant capabilities for the participants in focus groups, the use of an existing and robust work capability list to reflect on before the discussion, has turned out to be a decent method. Though one could also argue to create capability lists within each context or focus group first.

As the groups differed in the number of participants, this might have influenced the involvement of participants in the larger groups [33]. In the literature six to ten people per group has been found appropriate for discussion [34]. In the larger groups such as the HRM group, it was observed there was less dynamic discussion on personal conversion factors, whereas in smaller groups with workers that were more familiar to each other, such as the elderly care group, personal conversion factors were discussed more openly.

It is known that in focus groups different biases are faced, which might have impacted on the findings [35]. Participants tend to take many assumptions of others for granted much more easily and express culturally appropriate views only (acquaintance bias). In addition, there might have been a social desirability bias, towards colleagues and interviewers, as the meeting has been framed as a discussion on sustainable employability. Another possible factor impacting the findings in certain focus groups has been dominance bias (by which dominant individuals shape the conversations) and the halo bias (perceived status of participants influences the openness of sharing thoughts). Despite the Dutch horizontal consensus economy, employees higher in the hierarchy within the company might have naturally influenced the opinions and reflections of workers lower in the hierarchy during the focus group. Also, educational background and professional level of participants might have affected the depth of reflection on essential conversion factors by the group.

Our method has the limitation that conversion factors have only been examined at one point in time. This focus group method might benefit from using repeated discussions to track changes in conversion factors longitudinally and in respect to altering context. Triangulation with other qualitative research methods to validate the identified conversion factors or deepen understanding of personal conversion factors could be participatory or interview studies with the same participants.

## Implications for practice and further research

The recommendation following from this study is that companies and their employees might increase their sustainable employability by customized intervening on all three conversion levels to increase capabilities of workers. Our claim is not that large scale interventions or individual coaching sessions are not able to impact value achievement of employees, but a more sensitive approach to all levels of conversion might work out even better. Following this line of thinking it is recommended for companies working on increasing sustainability employability to experiment with combined interventions based on conversion factors on two or, preferably, three levels to increase capabilities of the workers, with repeated evaluative cycles of work values and conversion factors. For example, the effect of investing in optimal social contacts might be increased by simultaneously having attention for real opportunities to realize group ideas.

This study identified conversion factors that impact value realization in work. Further research is necessary to build the methodology of identifying capabilities and conversion

factors at the work floor. In addition, the effect of a value-based intervention at the work floor seems necessary to evaluate the effect on the sustainable employability of employees. In a prospective research design, a focus on work capabilities and conversion factors could be added to a "regular" large-scale occupational intervention and tested against only the large-scale intervention. Such approach might show the possible benefit of an additional aim to increase capabilities of all workers in order to maximize their sustainable employability.

## Author Contributions

**Conceptualization:** Patricia A. J. van Casteren, Evelien P. M. Brouwers, Arno van Dam, Jac J. L. van der Klink.

**Data curation:** Jan Meerman, Patricia A. J. van Casteren, Jac J. L. van der Klink.

**Formal analysis:** Jan Meerman, Patricia A. J. van Casteren, Evelien P. M. Brouwers.

**Investigation:** Evelien P. M. Brouwers, Arno van Dam.

**Methodology:** Jan Meerman, Patricia A. J. van Casteren, Evelien P. M. Brouwers, Arno van Dam, Jac J. L. van der Klink.

**Project administration:** Patricia A. J. van Casteren.

**Software:** Jan Meerman.

**Supervision:** Evelien P. M. Brouwers, Arno van Dam, Jac J. L. van der Klink.

**Validation:** Jac J. L. van der Klink.

**Visualization:** Jan Meerman.

**Writing – original draft:** Jan Meerman.

**Writing – review & editing:** Jan Meerman, Patricia A. J. van Casteren, Evelien P. M. Brouwers, Arno van Dam, Jac J. L. van der Klink.

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
