## [Decision Letter · Decision Letter 0]

8 Jun 2022

PONE-D-21-34683A capability perspective on sustainable employability: a Dutch focus group study on organizational, work and personal conversion factors.PLOS ONE

Dear Dr. Meerman,

Thank you for submitting your manuscript to PLOS ONE. After careful consideration, we feel that it has merit but does not fully meet PLOS ONE’s publication criteria as it currently stands. Therefore, we invite you to submit a revised version of the manuscript that addresses the points raised during the review process.

The manuscript has been evaluated by three reviewers, and their comments are available below.

The reviewers have raised a number of overlapping concerns, primarily noting the need for additional clarity to the theoretical motivation for the study, as well as to the methodological details. 

Can you please address these in your revision? 

We look forward to receiving your revised manuscript.

Kind regards,

Avanti Dey, PhD

Staff Editor

PLOS ONE

Journal Requirements:

Reviewers' comments:

Reviewer's Responses to Questions

**Comments to the Author**

1. Is the manuscript technically sound, and do the data support the conclusions?

Reviewer #1: Yes

Reviewer #2: Yes

Reviewer #3: Partly

2. Has the statistical analysis been performed appropriately and rigorously? 

Reviewer #1: N/A

Reviewer #2: Yes

Reviewer #3: N/A

3. Have the authors made all data underlying the findings in their manuscript fully available?

Reviewer #1: Yes

Reviewer #2: Yes

Reviewer #3: No

4. Is the manuscript presented in an intelligible fashion and written in standard English?

Reviewer #1: Yes

Reviewer #2: Yes

Reviewer #3: Yes

5. Review Comments to the Author

Reviewer #1: 1. Introduction: it is of interest to note that Jahoda did her seminal studies in the 1950s and 1960s, hence, the statement that nowadays other values are important needs more careful consideration!

2. Introduction: I am not sure that JDR model has shifted from outcomes of interventions...I would argue that the JDR model is used most in observational studies rather than intervention evaluation.

3. Introduction: the central aim is interesting, but comes a bit out of the bleu. What is already known? I would invite the authors to expand a bit on the novelty of the current study.

4. Methods: i feel that the authors use the term outcome very loosely...eg line 173 (see also remark 2)

5. Methods: how many participants were included in total, were focus groups done per company (as some of these choices were done as part of the design of the study, please shortly mention this in the method section)?

6. Results: participants, any idea about the educational background?

7. Results: I suggest to include a summarizing table of the conversion factors at three levels.

8. Results: Sometimes the language is a bit confusing, eg line 327 you start with limitations, and then you continue with a positive remark. Thus, the word limit should really be influenced..eg line 346 you introduce the phrase 'to get the grid around', which is difficult to interpret.

9. Results: the social contacts should be explained more for their target (colleagues, supervisor, clients etc). Contacts with the the latter group can be, according to literature, both be positive and negative.

10. Results: line 396 is strange...you start with defining personal level, and then present between group (which by definition is not about individuals)

11. Results: definition problem: is lack of challenge in the job an individual attribute?

12. Discussion: I think that the discussion should be more balanced with respect to evaluation of the direction of the conversion factors, eg hierarchy is presented as a problem, and this will depend most likely on educational/professional level of the participants, and also on cultural context. Thus, I would suggest to identify factors rather than interpreting their direction.

13. Discussion: the section on agency starts with a theoretical explanation, which should either follow from the results, or be introduced in the methods.

14. The results are about social contacts, in the discussion this seems ot be interprted as social coherence, which is something else.

Reviewer #2: The reviewed paper presents an important and up-to-date issue of a capability perspective on sustainable employability. The discussed topic requires further scientific exploration.

The article brings sufficient contribution to the development of knowledge in the concerned area. The article closely corresponds to the topic specified in the title. In this article all the issues were discussed in an understandable manner.

The literature selection presents current articles, but some of the root articles are also implemented. I suggest supplementing the subject literature based on the latest publications from this area. The purpose of the article should be clearly defined.

The argumentation and contribution of the paper is on a good level. The conclusions reflect the research results and are linked to the rest of the paper.

Reviewer #3: Review for PONE-D-21-34683 A capability perspective on sustainable employability: a Dutch focus group study on organizational, work and personal conversion factors.

Thank you for the opportunity to review this interesting qualitative paper on the capability approach and sustainable employability. The paper deals with a generally important topic and offers some insights into elements that relate to it. In that sense, the paper is an interesting read. However, in my opinion, the current framing of the paper does not do a convincing job at testing a theory or contributing to theory building (even if it does suggest to do so). The evidence presented and the way it is connected to conclusions, existing evidence, and the theory it is supposed to contribute to are insufficiently convincing. I think this could be solved by implementing a better framing and connecting to the existing sustainable employability literature better. However, it is at the same time unclear to me whether the quality of the data is sufficient to make claims to allow for positioning the identified factors as conversion factors. In sum, although the paper is interesting, I do think the authors have work to do regarding its framing and clarity of the methodological approach and connecting finding to results and conclusions that the paper sets out to offer. Below I offer several comments that are intended to aid the authors in doing so. I hope these are helpful.

Introduction / theory

A major issue with the current paper is that it is focused on sustainable employability but only incorporates a very limited account of that literature. There have been marked developments after the emergence of the capability approach (see for an elaborate critical review specifically: https://doi.org/10.3390/su12166366). The capability approach has been heavily criticized in the literature and several other approaches (that are somewhat compatible with the capability approach) have emerged, but the present paper does not relate to that discussion in any way. Certainly there are ways out of that with the capability approach still having relevance in the context of sustainable employability, but to have a convincing paper that deals with the issue of sustainable employability I think the authors need to connect to the state of the art thinking on this concept.

The paper incorporates several theories and concepts from the fields of work and organizational to connect to sustainable employability. However, the paper does not do a thorough job at explaining these concepts and theories and connections are very loosely made (e.g., ‘positive psychology’ which is a somewhat hollow and broad term; ‘self-determination theory’ which is not explained, ‘house of workability’ which offers little structure in terms of modeling relationships and is also not very clearly explained, the ‘job demands resources model’ which is not explained, ‘positive employability outcomes’ is used but then covers things not traditionally associated as such, ‘person-job fit’ is highlighted but not properly explained). Consequently, the point of incorporating these theories / concepts is not clear to me and I am not convinced that readers who are not very familiar with these theories and concepts can easily follow the line of reasoning. Related to the previous point, if a better connection would be made to the sustainable employability literature, this haphazard discussion of several (already debated) concepts would not be necessary and could be dealt with more efficiently. On a somewhat also related note, the introduction paragraph where this takes place is rather lengthy.

The discussion on the capability approach is easy to follow, but seems a bit imprecise and diverging from original formulations in key papers proposing that approach to sustainable employability. I recommend a more precise discussion and handling of key concepts. There is also some arguable discrepancy between how resources are framed within the capability model and how capabilities are actually measured in the original capability set. Statements suggesting that the capability approach does not focus on resources are therefore rather debatable.

It is somewhat strange that the paper has a paragraph on ‘operationalizing the capability set’ because this has been done in a previous paper already that this paper draws on. With this framing of the paragraph, it is not clear to me as reader what I need to do with the ‘operationalizing’ idea in this context. Perhaps a different frame would help.

The paper states ‘Although the capabilities of Dutch employees are established’. I find that a rather bold and highly debatable statement. What does it even mean? How can such a claim be made on the basis of a single reference? In this same paragraph there is marked repetition of previously made points on conversion factors. The ‘therefore’ sentence after this paragraph is also oddly disconnected from the paragraph itself, and arguably its content does not follow from the preceding paragraph at all. This in itself would not be a big issue, but what is currently still missing in the paper is a clear motivation as to why we should care about this topic at all (what is the issue at hand, why is it so relevant to know about capabilities?) and what important research gap is covered (what do we really know about the capabilities, why don’t we know the specific thing this study investigates, and why is that important to know?). Consequently, the contribution of the study also remains rather vague; what do researchers in this area or society in general stand to gain from this paper? I think these elements should be articulated much more explicitly and convincingly than is currently the case.

Several conversion factors are mentioned, but there is no clear argumentation as to why these factors would function as such. An example would be ‘age’, why would that be such an important conversion factor? And how does that work? I think the paper would benefit from an example or a more convincing explanation of this to help the reader understand conversion factors better.

The paper selects people from different sectors, but the logic behind selecting those sectors is never presented. Why these, why not others?

In the methods section, the paper states ‘We chose focus groups as a methodology to identify conversion factors, because it seems reasonable to take a similar strategy in line with methodologies in the capability literature as stated above.’ This is an odd line of argumentation; one should choose the method / design because it is best suited to answer the research question, not because ‘others did this too’.

Is ‘Focus group interviews’ an accurate and commonly used term? I am not sure.

The build up of the sample is completely unclear after reading the methods section. Information on the sample is needed there.

How purposive was the constellation of each focus group? Was there a specific reason for a certain build-up, was this buildup adequate for having the discussion?

Is there a coding tree?

Is there a protocol for the focus groups available to get insight into questions / discussion topics used?

In the Results section, it is odd that the research question is stated again, unnecessary repetition.

The results section connects some interpretations to industries/sectors; how adequate is the sampling approach for doing that? It doesn’t seem to me like the sample is built up in such a way that such conclusions are possible.

How convinced are the researchers that the ‘conversion factors’ listed are really ‘conversion factors’ because to me they are known aspects that make a work situation nice or horrible. What evidence do we have for the framing of conversion factors? The quotes used don’t seem to explicitly go into that direction (i.e., taking the idea that conversion factors represent the situation of a resource being available to an extent and then the things listed as conversion factors determine the extent to which that resource can be used to realize a certain value).

What is the ‘work level’? It seems more like ‘group’ / ‘team’ level factors discussed there .When I think of the work level, I would think of the nature of work and tasks that individuals perform (either alone or in teams).

The discussion section talks about agency, but where in the data do we see evidence of ‘agency’. Current write up is not convincing here because the connection is insufficiently explict. And relatedly, how is this different from the vast knowledge on autonomy?

What is social coherence? It seems like a complex concept but it is not clearly defined where it is discussed. And then, what is the conclusion here based on specifically.

The JDR paragraph is somewhat too limited. It offers an important (and previously formulated) critique to the JDR (i.e., demands and resources are non-concepts, because whether something is a demand / resource depends on how a person sees it). It could make this point more clear and connect to previous literature on this idea.

‘valid predictors of sustainable employability’. Read the suggested literature on sustainable employability to see that such a statement cannot be convincingly made at all, unless there is a profound quantitative study that longitudinally connects aspects to development in sustainable employability indicators over time.

Social desirability, presentation bias, and rationalizations potentially strongly limit the validity of statements made by participants and consequently conclusion drawn from them. More so than aspects mentioned.

It is too me very unclear what this study adds to existing literature. Even after reading the conclusions I have the impression that we learned nothing new. Maybe this is false, but it too a large extent relates to the fact that things are not adequately framed and connected to evidence to allow for specific novel conclusions and insights regarding conversion factors within the capability approach.

Abstract is very clear and well-written, but it would be informative to know a bit more about the data source (e.g., how many people, how long the discussions, which sectors represented).

6. PLOS authors have the option to publish the peer review history of their article (what does this mean?). If published, this will include your full peer review and any attached files.

Reviewer #1: No

Reviewer #2: No

Reviewer #3: No

---

## [Author Response · Author response to Decision Letter 0]

16 Jul 2022

I have addressed the additional requirements (e.g. style requirements, cover letter, references, other formatting samples) in the manuscript. 

I have moved the ethics statement to the methods section. 

Our research office will support me to upload the raw transcripts in a public repository. Information will follow after publication. 

See the "response to reviewers letter" for detailed responses to the feedback of all three reviewers.

---

## [Editor Report · Decision Letter 1]

22 Aug 2022

A capability perspective on sustainable employability: a Dutch focus group study on organizational, work and personal conversion factors.

PONE-D-21-34683R1

Dear Dr. Jan Meerman,

We’re pleased to inform you that your manuscript has been judged scientifically suitable for publication and will be formally accepted for publication once it meets all outstanding technical requirements.

Kind regards,

Sylwia Wiśniewska

Guest Editor

PLOS ONE
---

## [Editor Report · Acceptance letter]

13 Oct 2022

PONE-D-21-34683R1 

A capability perspective on sustainable employability: a Dutch focus group study on organizational, work and personal conversion factors. 

Dear Dr. Meerman:

I'm pleased to inform you that your manuscript has been deemed suitable for publication in PLOS ONE. Congratulations! Your manuscript is now with our production department. 

Kind regards, 

on behalf of

Ph.D. Sylwia Wiśniewska 

Guest Editor

PLOS ONE